# Enhancing Maritime Domain Awareness Through AI-Enabled Acoustic Buoys for Real-Time Detection and Tracking of Fast-Moving Vessels

**DOI:** 10.3390/s25061930

**Published:** 2025-03-20

**Authors:** Jeremy Karst, Robert McGurrin, Kimberly Gavin, Joseph Luttrell, William Rippy, Robert Coniglione, Jason McKenna, Ralf Riedel

**Affiliations:** 1Roger F. Wicker Center for Ocean Enterprise, The University of Southern Mississippi, Gulfport, MS 39501, USA; jeremy.karst@usm.edu (J.K.); joseph.luttrell@usm.edu (J.L.); william.rippy@usm.edu (W.R.); robert.coniglione@usm.edu (R.C.);; 2BLUEiQ, 10 Fan Pier Boulevard, Boston, MA 02210, USA; rmcgurrin@blueiq.us (R.M.); kgavin@blueiq.us (K.G.)

**Keywords:** acoustic target recognition, marine domain awareness (MDA), OpenEar™, artificial intelligence, localization

## Abstract

Acoustic target recognition has always played a central role in marine sensing. Traditional signal processing techniques that have been used for target recognition have shown limitations in accuracy, particularly with commodity hardware. To address such limitations, we present the results of our experiments to assess the capabilities of AI-enabled acoustic buoys using OpenEar™, a commercial, off-the-shelf, software-defined hydrophone sensor, for detecting and tracking fast-moving vessels. We used a triangular sparse sensor network to investigate techniques necessary to estimate the detection, classification, localization, and tracking of boats transiting through the network. Emphasis was placed on evaluating the sensor’s operational detection range and feasibility of onboard AI for cloud-based data fusion. Results indicated effectiveness for enhancing maritime domain awareness and gaining insight into illegal, unreported, and unregulated activities. Additionally, this study provides a framework for scaling autonomous sensor networks to support persistent maritime surveillance.

## 1. Introduction

Marine domain awareness (MDA) is essential for ensuring maritime security, tracking vessel activity, and protecting biodiversity from threats such as illegal, unreported, and unregulated (IUU) fishing [1,2,3,4,5]. Traditional monitoring systems like radar and optical sensors, while effective in many contexts, face significant limitations. Adverse weather conditions and line-of-sight constraints can hinder their performance, leaving critical gaps in surveillance. Acoustic sensing offers a persistent, weather-independent alternative, particularly when paired with advanced data analytics and artificial intelligence (AI) to enable the real-time detection, classification, localization, and tracking (DCLT) of vessel activity [3,6,7].

AI-driven solutions bring transformative capabilities to MDA by automating complex tasks traditionally requiring manual monitoring and adjustment [8]. Acoustic signals from vessels exhibit distinct patterns based on size, engine type, and speed [1,2,9]. While traditional signal processing methods demand extensive manual tuning and suffer from limited adaptability [10], AI models excel in analyzing large datasets, identifying patterns, and adapting to new conditions over time [11,12]. By operating at the edge, these systems provide real-time insights, significantly reducing communication demands and operational costs, while improving scalability. Moreover, the integration of AI enhances the precision of vessel tracking, even under challenging conditions such as dense shipping lanes, coastal regions with heavy traffic, and low-visibility scenarios.

Despite these advantages, the deployment of real-time AI for acoustic sensing presents challenges, particularly in power-constrained environments where frequent battery replacement or large solar arrays may not be feasible [13]. This study addresses these limitations by introducing BLUEiQ’s OpenEar™ system, a low-power, software-defined acoustic sensor platform that integrates AI models directly into the sensor. OpenEar™ enables autonomous and persistent surveillance, minimizing the need for continuous connectivity to cloud services while maintaining high levels of detection and classification accuracy. By processing data at the edge, the system achieves rapid response times and reduces the bandwidth required for data transmission.

This research evaluates the effectiveness of AI-enabled acoustic buoys equipped with the OpenEar™ system for detecting and tracking fast-moving vessels. This paper presents a qualitative analysis of an alternative approach to more exquisite DCLT solutions, such as towed arrays deployed from autonomous vessels. Traditional DCLT implementations, particularly those relying on towed arrays, offer high-fidelity acoustic capabilities, but they come with significant power, operational, and maintenance costs, stemming from the complexity of deploying and managing long, flexible sensor arrays that require precise handling, active stabilization, and dedicated platforms for operation [14]. By exploring a range of metrics, including detection range, localization accuracy, and the feasibility of onboard AI for enhancing MDA and deploying OpenEar™ sensors in real-world conditions, this work not only validates performance but also provides a scalable framework for deploying autonomous sensor networks to support persistent maritime surveillance. The findings aim to demonstrate how innovations in low-power AI and acoustic sensing can transform maritime monitoring, advancing both security and environmental conservation objectives.

## 2. Materials and Methods

### 2.1. Equipment Description

The OpenEar™ buoy system (BLUEiQ LLC, Boston, MA, USA) is a low-power, AI-enabled acoustic sensing platform designed for autonomous maritime domain awareness and scalable deployment, balancing cost-effective hardware with modular infrastructure to support larger networks. Scaling deployments with the OpenEar™ buoy system requires robust communication networks (satellite, LTE, or mesh), cloud computing infrastructure for sensor fusion, and integration with maritime situational awareness displays.

The system is designed for robust ocean deployment using two 30-inch life rings with a 17-inch inner diameter for flotation and a durable high-density polyethylene deck for structural support. Providing a buoyant force of over 200 N, the system encloses its electronics in a watertight case for protection. Secured to a cinderblock anchor via a rope, the buoy was previously tested for four days in the Gulf of Mexico in 100 feet of water, enduring rough seas and high winds. It withstood prolonged submersion and was dragged over a mile by strong currents, demonstrating its resilience in extreme maritime environments.

Each buoy is equipped with a hydrophone and preamplifier featuring a sensitivity of −160 dB re 1 V/μPa, a frequency response range of 10 Hz to 200 kHz, and a self-noise level below sea state zero across its operational bandwidth. This provides a high dynamic range (~120 dB) for detecting and classifying vessel acoustic signatures. The OpenEar™ Gen1 processing unit consists of an off-the-shelf USB sound interface sampling at 48 kHz with 24-bit samples and a low-power single-board computer. The Gen1 OpenEar™ buoy consumes up to 2 watts of processing power. The algorithms and model used on Gen1 have been benchmarked for an ARM Cortex-M4 processor with dedicated neural network hardware, allowing for real-time AI inference measured in tens of milliwatts. An ultra-low-power approach like this, compared with existing solutions using Jetson Nano processors consuming 10 to 15 watts, enables continuous monitoring without the need for frequent battery recharges or large solar arrays. The OpenEar™ system operates with onboard SD storage to retain raw and processed acoustic data, integrates a low-bandwidth LTE uplink for remote data transmission, and integrates a GPS receiver for position data and PPS time sync. Acoustic data are processed in real time in sliding windows of 1 s. The mean detection latency from detection to display has been measured to be <3 s depending on the network cloud latency.

By leveraging event-driven data transmission, OpenEar™ sends only short, relevant data clips to the cloud for sensor fusion and localization when a detection event occurs, significantly reducing bandwidth consumption while maintaining real-time situational awareness. Once in the cloud, detection data from multiple buoys are fused, allowing for vessel localization using time difference of arrival (TDOA) methods. This approach refines vessel tracking accuracy while minimizing computational load at the edge. The processed information is then made available via a real-time user interface, allowing maritime security agencies, including DHS and the Coast Guard, to access actionable intelligence for fast-mover detection and broader maritime surveillance applications. For this study and the development of the TDOA algorithms, the data from the buoys were postprocessed.

To ensure the accuracy and reliability of vessel detections, ground-truth verification was established by recording the GPS tracks of a test boat as it transited the test range. The test vessel’s GPS data provided a precise reference for validating detection events, ensuring that OpenEar™ correctly identified vessel presence, location, and movement. The system’s performance was assessed by comparing the time-stamped detections with the known GPS-reported positions of the test vessel. Additionally, detections were analyzed for consistency in localization accuracy and signal characteristics to ensure the robustness of the TDOA-based approach.

### 2.2. Data Collection

Three BLUEiQ OpenEar™ sensors were deployed off Gulfport, MS, on 31 July 2024, to evaluate their performance under real-world conditions for vessel detection and localization. The test involved collecting data from three OpenEar™ sensors to analyze TDOA measurements and localization accuracy. Data collection specifically targeted Go-Fast boats, historically associated with illicit activities like drug or contraband transportation due to their speed and stealth capabilities. To mimic these scenarios, the test utilized the 26’ Avenger fishing boat “Legends of Lower Marsh”, equipped with a 300 HP Yamaha engine capable of speeds up to 45 knots. The evaluation also aimed to determine OpenEar™’s real-time vessel detection capabilities as well as DCLT using its recorded audio. Boat passes were conducted at varying speeds, at distances as high as 1.5 km, and in different directions (north, south, east, and west) in 29 °C waters of less than 4 m, with OpenEar™’s hydrophones deployed at a depth of 2 m (Figure 1 and Figure 2). The sensors were configured in a roughly equilateral triangle with ~100 m on each side and remained in the water for six days. Data were downloaded at the end of collection for processing and analysis.

Data collection comprised ground-truthing fishing boats as they passed the buoy and hand-labeling the collected data for accuracy. Approximately half of the dataset consisted of fishing boat activity, while the other half captured ambient conditions, ensuring a balanced representation of scenarios for training the model. This structured approach provided a robust foundation of labeled data for building and testing the vessel detection algorithm.

To minimize localization errors, buoys equipped with acoustic sensors need be optimally deployed according to research objectives and environmental conditions. The optimal deployment of the acoustic AI buoys consisted of positioning the sensors in a triangular formation to maximize coverage and, thus, localization accuracy.

### 2.3. Data Processing

The performance of passive acoustic monitoring in shallow-water environments is highly influenced by environmental conditions, which were measured and analyzed during this study. Ambient noise levels varied due to biological activity, wind-driven surface noise, and anthropogenic sources, all of which influenced the detection range and classification accuracy. Shallow-water reverberation increased multipath interference. These environmental factors were incorporated into the system’s AI training and calibration, improving the robustness of AI-driven vessel classification and localization.

To estimate localization, a three-step data processing pipeline was developed for this study. The steps for the localization and signal processing pipeline included the following:Acoustic Signal Capture: this was needed because OpenEar™ sensors capture sounds across a wide frequency band (10 Hz–200 kHz), which is critical for identifying and distinguishing various vessel types.Feature Extraction: features such as spectral peaks and time-domain patterns were extracted from the signals using Mel-Frequency Cepstral Coefficients (MFCCs).Classification: AI models, trained on pre-collected data, classified vessel types based on their acoustic signatures. For example, fast boats produced distinct high-frequency signatures, which the trained model could identify in real time.

The raw data collected during the test were processed into more manageable data files. Processing data consisted of organizing data into a usable format for analysis (3 wav files, one for each sensor) and a single JSON file containing the vessel and sensor location data for the duration of the event. The sound data would be for DCLT using both AI and signal processing techniques, with the JSON data as the ground truth.

### 2.4. AI Model

A vessel detection model was developed based on the data collection described above. The vessel detection AI model developed by BLUEiQ is based on a lightweight convolutional neural network (CNN) optimized for real-time acoustic classification on low-power edge hardware. The architecture consists of four convolutional layers designed to extract spectral and temporal patterns from processed acoustic data, allowing the model to recognize vessel signatures with high accuracy. These layers are followed by two max-pooling layers, which reduce dimensionality and enhance feature robustness while preserving essential information. After feature extraction, the model includes one fully connected layer, which aggregates the learned features to make meaningful predictions. Finally, a softmax output layer converts these features into vessel class probabilities, enabling the real-time classification of acoustic events. This structured approach ensures that the model efficiently captures relevant patterns in vessel acoustics while remaining computationally efficient. To maintain efficiency on resource-constrained devices, the model is designed with approximately 500,000 parameters, carefully optimized for low-power edge inference. The total model size is less than 1 MB, allowing for seamless deployment on OpenEar hardware without excessive memory or storage requirements.

To develop the model, BLUEiQ leveraged Edge Impulse, a web-based AI development platform, to streamline the process of training and optimizing the AI for the Gen1 OpenEar™ hardware. By feeding the collected and labeled data into the platform, the team designed and trained a real-time AI model tailored to the constraints of their low-power edge device using TensorFlow Lite v2.16.1. The resulting model, optimized using Edge Impulse tools, was successfully deployed on the OpenEar™ platform, achieving accurate real-time detection and classification of vessels during subsequent deployments. This iterative development process enabled BLUEiQ to efficiently refine their AI solutions while adhering to the power and performance requirements of their hardware.

The model trained on nearly six hours of recorded data clips, including various high-speed boats captured outside the Port of Gulfport. Data collection took place during two separate events in Gulfport, in March and June of 2024. The dataset was carefully structured, incorporating vessel recordings under diverse conditions—varying in range, direction, and ambient noise—as well as ambient-only clips. To ensure balanced training, the positive (vessel) and negative (ambient-only) classes were split nearly evenly. The trained model achieved high performance metrics, with a precision of 98%, a recall of 98%, and an F1 score of 98%, demonstrating its robustness in detecting vessel presence across different acoustic conditions. Similarly, the model was capable of real-time inference with a latency of 4 ms using less than 130 K RAM, ensuring that vessel detection was processed near-instantly on low-resource devices. The TensorFlow Lite model was output from Edge Impulse and integrated with the OpenEar™ sensor application and ran in real time.

The vessel detection AI model utilized MFCCs as input features for a 2D convolutional neural network (CNN). MFCCs, derived from audio signals, offer a compact and perceptually aligned representation of spectral properties that mimic human auditory perception. The data preprocessing pipeline specifically for model input included applying a pre-emphasis filter to amplify high-frequency components, segmenting the audio into overlapping time windows, and performing a Fast Fourier Transform (FFT) to convert the signal into the frequency domain. The frequency spectrum was then mapped to the Mel scale and transformed into logarithmic form to mimic the human ear’s sensitivity to loudness. Finally, a Discrete Cosine Transform (DCT) was applied to produce the MFCCs, which were organized into a 2D matrix resembling a spectrogram for CNN processing.

The CNN architecture captured temporal and spectral features from the MFCC matrix, leveraging convolutional layers to detect localized patterns such as harmonics and frequency transitions. Max-pooling layers reduced dimensionality while preserving prominent features, adding translation invariance for robustness. The use of ReLU activations enabled the model to capture non-linear relationships, distinguishing subtle variations in audio signals. Fully connected layers aggregated the learned features for decision-making, while the final softmax layer converted these into probabilities for classification.

During training, hyperparameters were optimized to enhance model performance. A learning rate of 0.01 was employed alongside the Adam optimization algorithm to ensure efficient convergence. Regularization techniques included batch normalization to improve generalization and mitigate overfitting. The model required 500 epochs to achieve training stabilization. Additionally, the MFCC window size was fine-tuned to optimize feature extraction for robust acoustic classification. This approach, combining the strengths of MFCCs and CNNs, proved well-suited for low-power edge AI applications like the BLUEiQ OpenEar™ system, where efficiency, robustness to noise, and adaptability to environmental variability are critical.

### 2.5. Estimating Location

Several localization methods [15,16,17] that were not used in this study were ruled out due to the large distance between microphones causing a lack of phase-coherent wavefronts at high frequencies. Excluded methods included Direction of Arrival [18] and Beamforming methods [19,20,21], as well as hybrid approaches [17]. The TDOA [22] method with Generalized Cross-Correlation with Phase Transform (GCC-PHAT) was chosen due to its ability to selectively correlate specified frequency bands of interest, its noise-robust performance, and its ability to locate non-phase-coherent audio sources.

Incoming audio data were first processed by splitting time-aligned audio samples from the sensors into segments. Segment size was chosen to ensure the inclusion of maximum possible time differences between sensors and to provide significant spectral context for correlation. Segment length was also chosen to exclude significant amounts of motion of the target. Test sweeps of this parameter were made and a segment length of 0.5 s with a 0.25 s overlap between segments was found to best correlate with our ground-truth GPS measurements of target data for our test cases.

In this work, the TDOA between microphone pairs was estimated by taking the phase transform of the audio sources using a traditional FFT modified with a Hanning window to minimize artifacts. The phase space arrays were then used to calculate a cross-power spectrum by multiplying one with the conjugate of the other, and the resulting spectrum was weighted with a multiplier in a frequency band of interest. The cross-correlation between the phase arrays was extracted by taking the inverse FFT of the weighted cross-power spectrum, and then the most likely TDOA (τ) was calculated by finding the highest peak within a maximum possible time delay calculated from the distance between microphone pairs and reasonable assumptions of the speed of sound. A factor of 10% was added to this estimate to account for possible errors in position measurement between microphones (GPS-derived) or errors in our assumptions about the local speed of sound. A short-window moving average was applied to the correlation result to ensure the absence of high-frequency noise, potentially interfering with peak detection. The TDOA was calculated to be the time offset of the peak of the correlation.

After the TDOA calculation, we used a confidence metric to estimate the quality of the TDOA measurements. This metric was calculated as follows: We located the second highest peak of the autocorrelation, that is, at least one zero-crossing away from the main peak. The signal noise floor was estimated to be the 95th percentile of the autocorrelation. Percentile was chosen rather than standard deviation for the noise floor estimates due to the non-Gaussian distribution of the correlation and due to observed improved confidence metrics in a-b testing. The confidence metric was constrained to be a value between zero and one, calculated as 1 − ((second peak − noise floor)/(highest peak − noise floor)). The confidence metric was set to zero if either the correlation index peak corresponded to an unrealistic TDOA as per microphone distances, or if no second peak in the correlation was found (rare edge case).

The above GCC-PHAT implementation was chosen due to its higher performance in noisy or reverberant environments, its reduced sensitivity to narrow-frequency noise, and its increased accuracy in TDOA detection and because it allowed for a robust metric to be used to gauge estimation confidence. Many variations of this algorithm, filtering steps, and confidence metrics were explored during this study. A plot for visualizing the results of the GCC-PHAT correlation and corresponding TDOA measurements for a simulated example case are shown in Figure 3 below.

Once TDOA estimates were made, a system of equations could be solved to calculate a maximum-likelihood position for the target by solving the intersection of hyperbolic curves implied by the TDOAs between pairs of sensors. We could greatly simplify our solution space by making several key assumptions: that there was a single dominant source within the operational area of the sensors, that this source and all sensors laid within a plane, and that the positions of the sensors and local speed of sounds remained relatively constant. Given these assumptions, the TDOA equations simplified to a set for which an approximate solution could be more easily found.$$D_{0}\;=\;\sqrt{{\left(x-M_{0.x}\right)}^{2}+{\left(y-M_{0.y}\right)}^{2}}$$$$D_{1}\;=\;\sqrt{{\left(x-M_{1.x}\right)}^{2}+{\left(y-M_{1.y}\right)}^{2}}$$$$D_{1}\;=\;\sqrt{{\left(x-M_{1.x}\right)}^{2}+{\left(y-M_{1.y}\right)}^{2}}$$$$D_{0}\;-D_{1}={TDOA}_{0-1}*C\;$$$$D_{1}\;-D_{2}={TDOA}_{1-2}*C$$
*D*_0_: sensor zero;*D*_1_: sensor one;*D*_2_: sensor two;x, y: estimated target location in local coordinates;$M_{i.x}$: microphone X position for microphone index *i*;${TDOA}_{i\text{-}j}$: time difference of arrival between microphones *i-j*;C: speed of sound in water.

An optimization routine was used in this study to solve the TDOA equations above for the hyperbola pairs between microphone triplets. Traditional methods of optimization for localization [23] including all microphone pairs and their corresponding TDOAs were not used in this study because of their higher complexity, resulting in a wider non-linear solution space, making it computationally prohibitive to find a solution. Instead, the equivalent hyperbola for each microphone pair was calculated discreetly, and the intersection for each pair of hyperbolas was estimated by using the scipy v1.15.0.fsolve module from Python v3.11. The optimization routine was initialized with several guesses which should have resided within each of the possible local minima of the solution space (at the midpoint between microphones and in each direction perpendicular to the vector between microphones). Note that this approach is unsuitable for an array with many microphones due to its complexity. The implied hyperbolas from the previous example signal and its TDOA estimates are shown in Figure 4 below.

The intersections of these pairs of hyperbolas were used to calculate a corresponding range and bearing from the centroid of the microphone array, and the median angle was used to choose an intersection to represent our target location. Angle was used as the deciding factor for choosing a target intersection because the TDOA is far more accurate in predicting angle than range. Range was calculated from this chosen intersection and bounded to a maximum to prevent issues near hyperbola asymptotes. A confidence metric was assigned to this target location, calculated as the minimum of the confidence of all the TDOA pairs. The minimum was used because localization required all three TDOAs, and errors in any one of them resulted in errors in the final location.

### 2.6. Target Position Filtering

Position filtering was necessary following TDOA estimation to remove outliers and consolidate a likely target path over time. Using a long-window moving average, the region of the highest detection confidence was estimated by taking the largest contiguous region above a minimum confidence threshold. The resulting filtered region removed unrealistic data corresponding to points too distant from sensors to be reliable.

Within the high-confidence filtered region, a noise-robust covariance filter was used to find the likelihood of a datum being an outlier. The likelihood was based on determining the Mahalanobis distance [24] for a TDOA triplet. The Mahalanobis distance was determined based on a sliding window of 21 samples over all TDOA triplets. Each sample corresponded to a 0.5 s audio segment with 50% overlap between segments. If a given triplet was found to be statistically unlikely (an outlier), it was replaced with the mean in the TDOA 3-space for that window. The above covariance filter used the support fraction and estimated percentage of outliers as inputs. After calculating the covariance filter, the weighted moving average of the filtered was used to collapse the point cloud into a line of highest probability.

The highest-probability TDOAs were used to generate the local X and Y coordinates of the target. An estimated closest approach was calculated from a long-window smoothed distance obtained from the unfiltered local coordinates, used to bifurcate the position data. The first section of these data was reversed to allow the application of a Kalman filter. Data bifurcation and reversal before the Kalman filter was due to the stateful nature of the filter. The Kalman filter allowed for state initialization and filtering data from the most to the least reliable (nearest to farthest). This was performed to minimize the impact of less reliable data on the position estimates when the target was near the sensors.

The bifurcated and reversed predicted XY coordinates were passed through a purpose-built Kalman filter. The custom filter was stateful, which used information about the maximum likely acceleration of the vehicle. The filter was also based on positions being more uncertain along the distance than along the angle and on how the sensed location was more accurate in its predicted bearing than range. This method made predictions close to the origin more accurate but could deteriorate with distance due to vessel acceleration and motion occurring far from the origin, where distance estimates were poor. After filtering was completed, the first section of the data was reversed to its original time-ordering and rejoined with the filtered second section.

## 3. Results and Discussion

Data collection using the three OpenEar™ sensors enabled effective TDOA measurement and localization analysis. The sensors captured data from multiple angles, allowing for a thorough evaluation of system performance in noisy shallow-water conditions. Despite test ranges being limited to under 1.5 km, real-time vessel detection was validated, with the system successfully identifying vessels during all 16 test events. The inclusion of boat passes at varying speeds, distances, and directions further demonstrated OpenEar™’s capabilities in diverse and realistic scenarios, showcasing its potential for vessel detection and localization in shallow-water environments. Vessel distance to the array center was clearly shown when real-time vessel detections occurred for each event (Figure 5). Algorithm tuning to decrease the minimum sound pressure level threshold resulted in improved performance, with increased detection from 750 m to 1300 m, near the maximum distance of the test range (Figure 6).

The detection of vessels operating in shallow waters is a challenge due to the higher sound reverberation in such environments and the diversity of soundscapes from biological sources [25]. Shallow waters, however, are also the epicenter of human maritime activity, including vessel transit to ports, marinas, and passage channels, where reliable detection is essential [26,27]. The TDOA detection approach of this study produced detection ranges extending past 1 Km in shallow water, enabling the development of an effective AI-based system recognizing and describing vessel behavior.

The vessel detection AI ran in real-time on OpenEar, identifying signals of interest directly on the platform with minimal human intervention. Once detected, these signals could be transmitted from all buoys in the array and aggregated in the cloud, where time difference of arrival (TDOA) calculations were performed to extract detailed information on the source’s heading and position over time. In the following analysis, the first of the sixteen test events from our dataset is presented, where a vessel approached the sensor array from several hundred meters away, passed between the sensors, and traveled away at a relatively constant speed (Figure 7). The analysis indicates the interference of vessel signals from background, environmental noise. Despite interference, vessel noise was still apparent, demonstrating the utility of our system even in sub-optimal locations, such as nearshore habitats.

Despite the three sensor platforms consisting of the same hardware, with hydrophones at similar depths, and with roughly 100 m of separation, the characteristics of their acoustic waveforms in the far-left column of the subplots are noticeably different (Figure 7). In the test location, a variety of factors may have accounted for these differences, including biological factors like snapping shrimp (*Alpheidae* sp.), variations in bottom depth and composition, and distance to other waterborne noise [25]. Figure 7 (column 2; A2–C2) show the expected falloff of higher-frequency components of the noise source at larger distances, in addition to strong-broadband and higher-frequency components of the vessel signature as it neared and passed between the sensors. Moreover, there is a strong indication in Figure 7(A3–C3) of biological activity, which manifests as vertical broadband lines at irregular intervals. Biological noise was prominent even in our frequency bands of interest for vessel detection and was the driving force for several filtering methods and algorithmic improvements made during the development of the localization techniques in this research. Biological sources caused correlation in the GCC-PHAT step of TDOA estimation to contain large secondary peaks, masking the primary peak in some audio segments.

The relative confidence value for a given TDOA estimate was visualized by modulating the transparency value of each point (Figure 8). A floor value was applied to the transparency values to visualize very-low-confidence estimates. Similarly, a ceiling value was applied to ensure the visualization of nearly overlapping estimates, including estimates of very high confidence. These two plotting choices tended to qualitatively overrepresent extremely-low-confidence values relative to high-confidence ones.

The initial TDOA estimation for this study’s test agreed well with the expectation of cases where the vessel was within a few hundred meters of the sensor array. As the distance increased, however, the TDOA value and associated confidence became less reliable, represented by the transparency value of data points (Figure 8). An additional feature of the TDOA estimates is that they also became less reliable as the vessel was extremely close to, or between, the sensors. This is assumed to be influenced by the broadband noise becoming more dominant and less structured as seen in the spectrograms of Figure 7. The loss of TDOA confidence at close proximity to sensors was also influenced by multipath effects becoming more significant with a larger ratio between the distance to the bottom and the distances to the sensors. Also, correlation between sensors was expected to be lower when a noise source was closer to only one sensor. This factor was due to the larger differences in the recorded noise signature as the vessel was much closer (as a ratio) to some sensors than to others, with the directionality of the noise source playing a larger role [28].

Using the initial TDOA results together with the hyperbolic intersection methods discussed above, the solution for the maximum-likelihood noise source position for each TDOA triplet was found. In Figure 9, we show these positions in local Cartesian coordinates, with (0, 0) as the centroid of the microphone array. The ground truth comprised the GPS data from the test vehicle converted to local coordinates. As for the TDOA results above, the X and Y position estimates strongly agreed with ground truth except for larger distances, as the vehicle became too distant for reliable position estimation, and for close distances, as the solutions were of low confidence.

The TDOA-based location had better angular than distance accuracy due to the amount of error in the timing estimate resulting in ever larger distance differences as the implied hyperbolas became closer to the asymptotes at their intersections [29]. For this reason, we also show the equivalent angular and distance estimates in subplot B, where we can see strong agreement in the angular measurements at modest ranges while the distance estimate gradually became worse (Figure 9).

After the filtering methods detailed in the previous section, a region of high confidence was chosen for localization. Within this region, the localization was assumed to be reliable inside the region where TDOA estimation became impractical due to distance. In addition to selecting a high-confidence region, filtering removed the vast majority of outlier TDOA measurements. For much of the region plotted, the TDOA measurements closely agreed with the GPS-derived ground truth (Figure 10). The various acoustic confounding factors for the region where the vessel was between sensors were limited to approximately 85–90 s. In this region, the TDOA estimates rapidly changed, reflecting vessel movement, and did not agree as strongly with the ground truth. To assist in visualization, a Kalman filter was used. After our Kalman filter was applied, the position estimates agreed with the ground truth within several hundred meters of the sensors and were still accurate enough beyond 600 m (Figure 11).

The process of generating localization results from audio samples in this study included over a dozen configured parameters, with additional assumptions about the target, such as the maximum acceleration, maximum detectable distance from the microphone array, and frequency band for the GCC-PHAT correlation. Each of the configured parameters was either selected manually using estimates of real-world conditions, such as the speed of sound in water, or was selected by running a parameter sweep across all test cases, prior to optimizing for agreement with the ground truth, emphasizing such agreement closer to the sensor array. Further refinements of these methods may involve writing an optimization routine and using simulated annealing or gradient descent to select parameters across a larger set.

## 4. Conclusions

In this study, we show that the integration of machine learning with low-power acoustic sensing systems like OpenEar™ provides a scalable and transformative solution for enhancing maritime domain awareness. By combining AI-powered DCLT capabilities running on commodity edge hardware, with flexible deployment strategies and cloud-based data refinement and analysis, OpenEar™ offers a significant logistic and cost advantage in combating illegal activities and protecting marine environments. The integration of AI not only automates data processing but also improves the efficiency and accuracy of monitoring systems, ensuring that maritime assets and ecosystems are safeguarded.

The TDOA-based detection approach demonstrated in this study achieved detection ranges exceeding 1 km in shallow water, overcoming many of the limitations typically associated with acoustic sensing in high-reverberation environments. This capability enabled the development of an AI-based system capable of recognizing and characterizing vessel behavior in complex maritime settings, improving situational awareness and operational effectiveness in nearshore environments.

This study primarily focused on shallow-water detection for fast movers; future work will expand to deep-water testing to validate the feasibility of IUU fishing detection in open-ocean environments. Planned deployments will assess OpenEar™’s ability to detect and classify vessels operating beyond the range of coastal surveillance assets. This will ensure that the same low-power, edge-based AI inference model used for fast-moving tracking can be adapted for long-range vessel monitoring in deep-sea environments. By integrating ultra-low-power AI inference with cloud-based fusion, OpenEar™ provides a scalable and cost-effective solution for persistent maritime domain awareness across multiple mission profiles. Moreover, OpenEar™ surveillance may serve as a first means for detection to be followed by resource-hungry detection systems deploying multiple sensors and buoys arranged in complex array configurations.

Limitations of the current AI approach must be acknowledged to provide directions for future work. Data collection faced several logistical and environmental challenges. Schedule and budget constraints limited testing to shallow waters with a restricted number of boats, preventing assessments in deeper offshore environments. Future research could expand deployments to deeper waters with additional vessels to improve the robustness of localization and classification models. Ambient noise sources also posed challenges, particularly high levels of background noise from the nearby Port of Gulfport and biological interference from snapping shrimp, which generated sounds reaching 155 dB re 1 µPa. Despite these conditions, the trained machine learning model demonstrated resilience, effectively distinguishing target acoustic signals from the noise. These findings highlight the robustness of OpenEar™’s AI-driven processing in real-world maritime environments.

In high-noise scenarios, such as areas alluded to above, with significant biological activity or heavy maritime traffic, the system may encounter challenges in distinguishing target signals from background noise, which may lead to reduced detection accuracy or false positives. In our application, the AI detection features are used primarily as a first filtering measure before moving data off-platform, and so some of these downsides are mitigated with further processing after the data are sent. Additionally, distance-based limitations arise as the localization accuracy diminishes with increased range from the sensor array. The ability to resolve precise positions is constrained by factors such as multipath propagation, phase-coherence issues, and environmental variability in shallow or complex underwater topography.

To address these challenges, future research may focus on improving noise-robust algorithms and incorporating advanced filtering techniques to mitigate the impact of high-frequency and ambient noise. Exploring adaptive AI models [30] capable of dynamic recalibration based on environmental conditions could enhance the system’s flexibility and accuracy. Expanding sensor networks to cover larger areas in deeper waters and optimizing their deployment configurations through simulation and field trials will further improve coverage and localization performance [31,32]. Additionally, integrating complementary data sources such as oceanographic and weather data may provide a holistic view, potentially enhancing the system’s ability to make accurate and context-aware predictions [32]. Moreover, future research may further explore the use of low-cost distributed sensors on attritable autonomous platforms to improve real-time localization, data fusion, and adaptive sensing strategies. Investigating optimized deployment strategies, multi-modal sensor integration, and advanced AI-driven processing for detecting and classifying maritime threats will be critical to advancing this paradigm. This direction will enable a transition from expensive, exquisite arrays to scalable, autonomous sensor networks, transforming MDA operations and expanding persistent monitoring capabilities across vast ocean regions. OpenEar™, as a distributed array, represents a significant advancement in MDA by enabling scalable, cost-effective acoustic sensing. Traditional approaches rely on high-cost, static, or limited-mobility sensor arrays, whereas OpenEar™’s integration with low-cost, attritable autonomous vessels provides a more flexible and persistent monitoring solution. By demonstrating real-time localization capabilities through distributed sensing, as outlined in this paper, OpenEar™ contributes to the shift toward autonomous, networked surveillance systems that enhance situational awareness while reducing deployment costs.

This study validated the effectiveness of OpenEar™ as an open platform capable of supporting real-time AI models developed with widely used tools like TensorFlow and PyTorch. The successful deployment of OpenEar™ systems for data collection, coupled with remote data access and model training using accessible AI frameworks, underscores the platform’s flexibility and adaptability. By demonstrating its ability to process real-time data for tasks such as vessel detection and classification on low-power hardware, this research highlights OpenEar™’s potential as a versatile solution for edge AI applications. The streamlined workflow—from data collection to real-time inference—enables users to develop and deploy custom models tailored to their unique requirements, leveraging industry-standard tools to address pressing maritime challenges. Moreover, the qualitative analysis in this paper underscores that while this alternative approach may not fully replicate the spatial coherence and sensitivity of long towed arrays, it offers a compelling tradeoff by enabling persistent, scalable, and cost-effective passive acoustic monitoring. The ability to deploy multiple small-scale systems with onboard AI processing allows for real-time decision-making, significantly improving maritime domain awareness while reducing logistical overhead. Ultimately, this paper positions this approach as a practical, scalable alternative to exquisite DCLT solutions, making advanced acoustic sensing more accessible for a broader range of defense, security, and environmental applications.

## Figures and Tables

**Figure 1 sensors-25-01930-f001:**
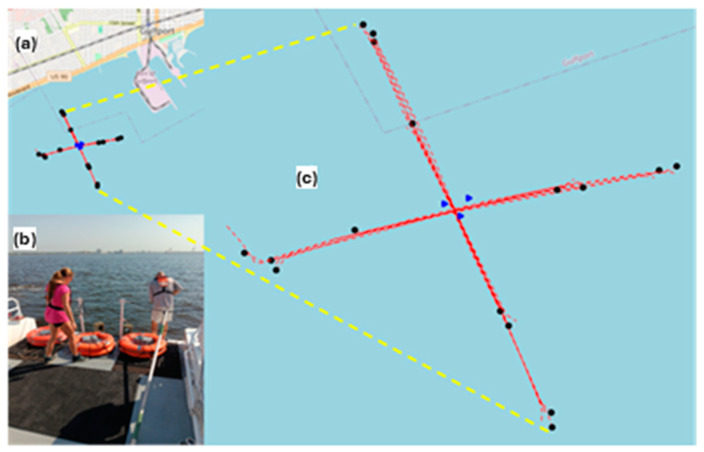
BlueIQ OpenEar™ sensor data collection area; the test boat path is shown by red dashed lines, the sensor locations by blue triangles, and the starting points for each test event by black circles. The test range is shown in (**a**),the deployment operation for the 3 OpenEar™ buoys in (**b**), and a zoomed view in (**c**).

**Figure 2 sensors-25-01930-f002:**
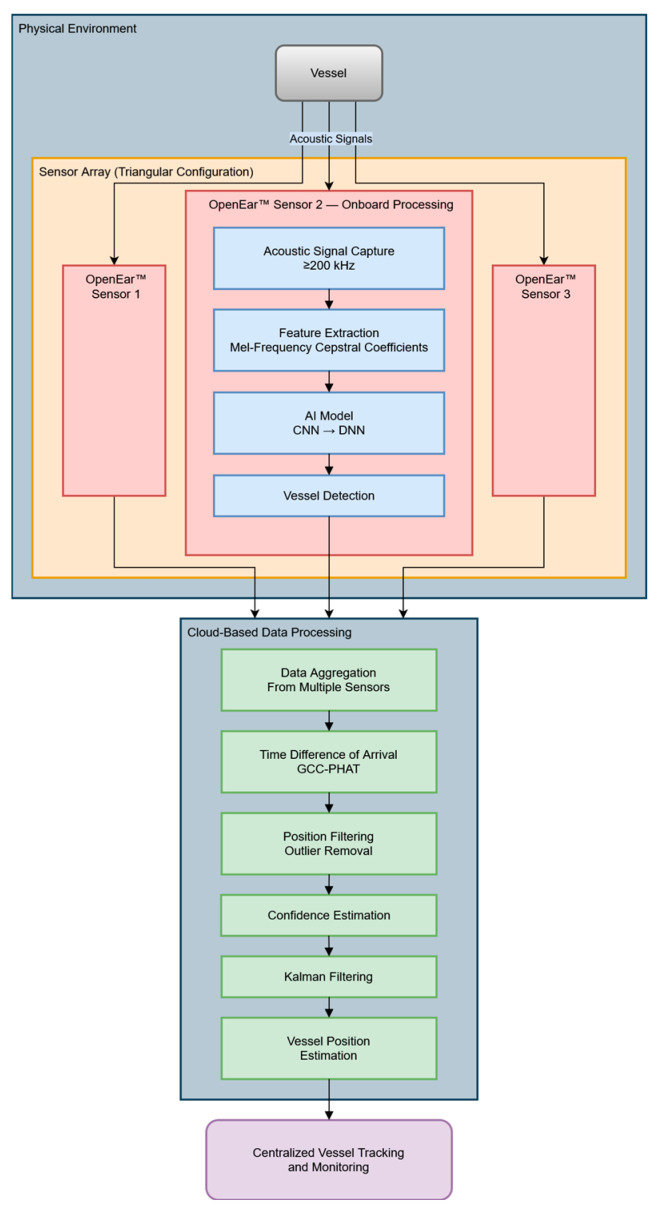
BlueIQ OpenEar™ sensor data capture, processing, and deployment pipeline.

**Figure 3 sensors-25-01930-f003:**
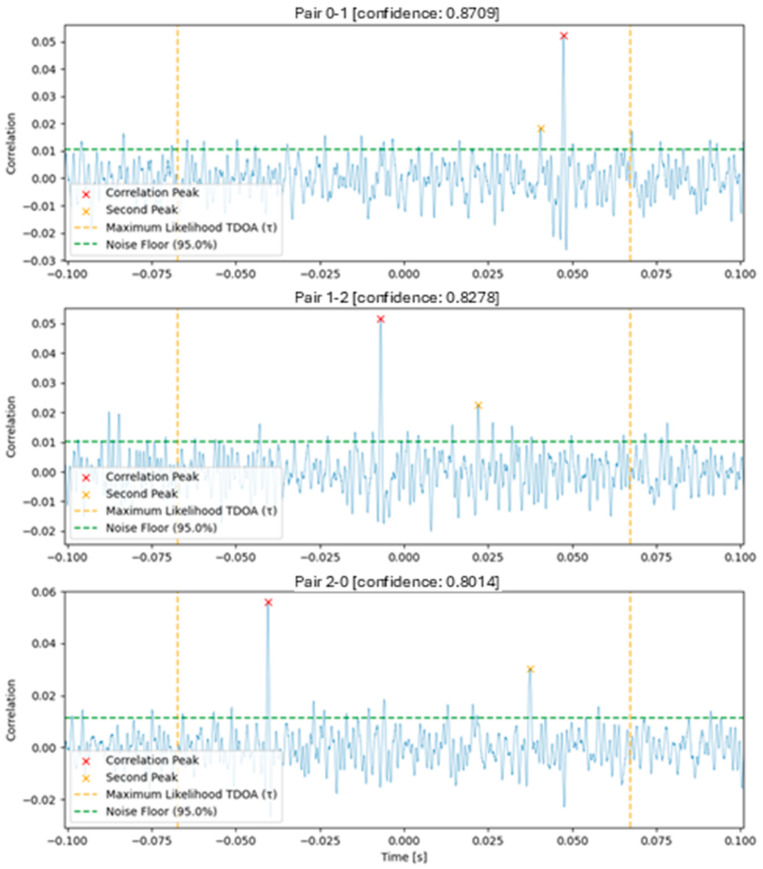
GCC-PHAT correlation from example signal between each pair of three microphones.

**Figure 4 sensors-25-01930-f004:**
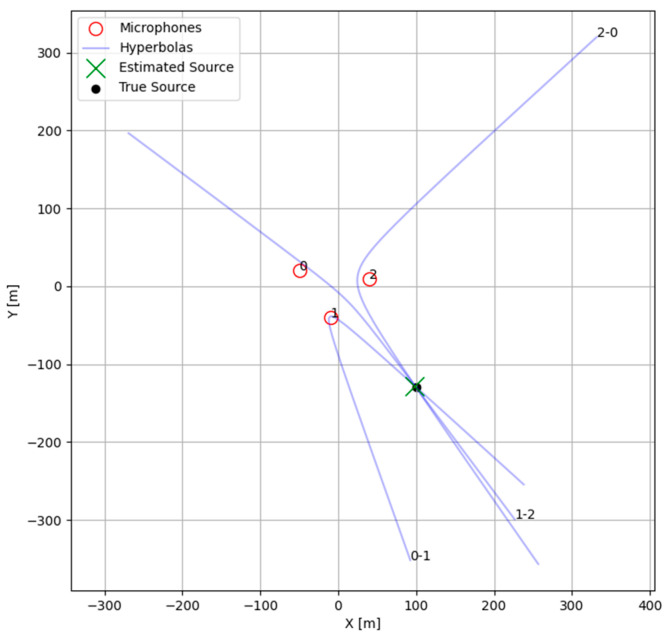
Localization estimate for an example signal using Generalized Cross-Correlation with Phase Transform.

**Figure 5 sensors-25-01930-f005:**
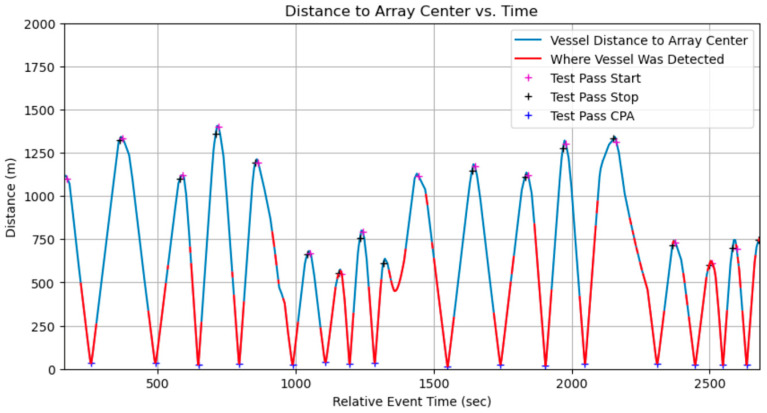
Real-time detection of vessel by AI model running on OpenEar™.

**Figure 6 sensors-25-01930-f006:**
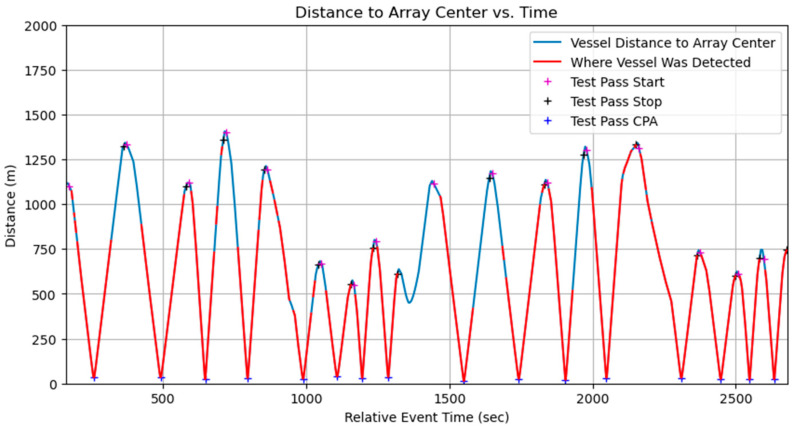
Reprocessed data after algorithm tuning showing extended detection ranges.

**Figure 7 sensors-25-01930-f007:**
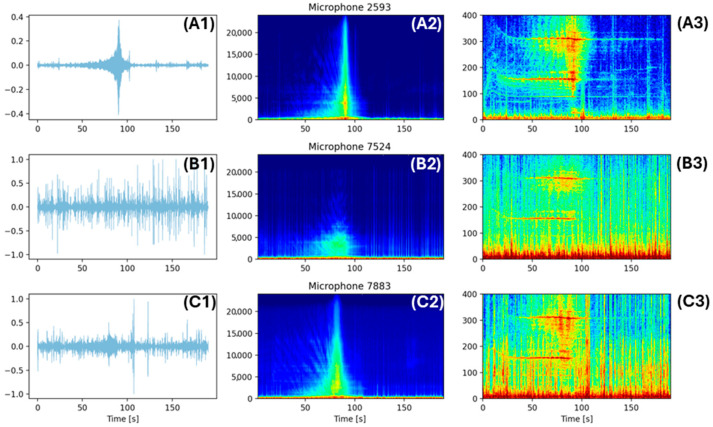
Acoustic waveforms and high-resolution spectrograms for watercraft; each row represents a different buoy microphone; blue are low intensity signals, red are low. Column 1 (**A1**,**B1**,**C1**) is the audio waveform showing a cleaner waveform for (**A1**), column 2 (**A2**,**B2**,**C2**) is a full-frequency spectrogram showing weaker frequency for microphone (**B2**), and column 3 (**A3**,**B3**,**C3**) shows a narrow-band, high-resolution spectrogram where microphone (**C3**) is noisiest.

**Figure 8 sensors-25-01930-f008:**
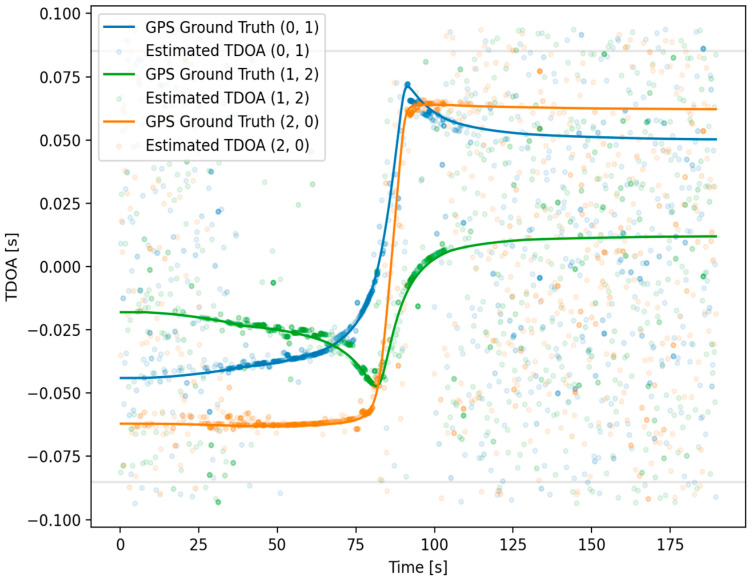
Initial TDOA estimation results.

**Figure 9 sensors-25-01930-f009:**
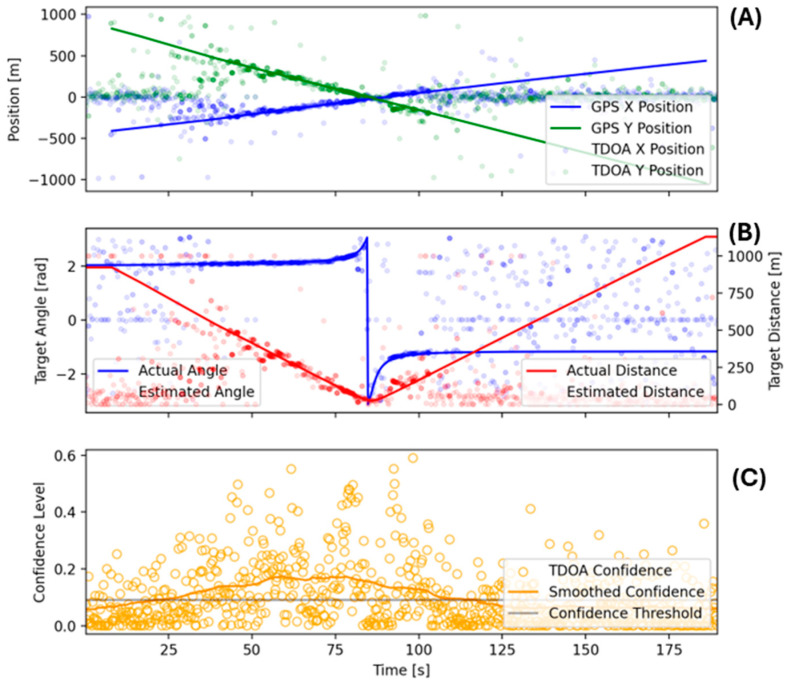
Initial position inferences from TDOA results; X and Y position of similar accuracy (**A**); actual and estimated distance show medium noise level (**B**); confidence shows most point within threshold (**C**).

**Figure 10 sensors-25-01930-f010:**
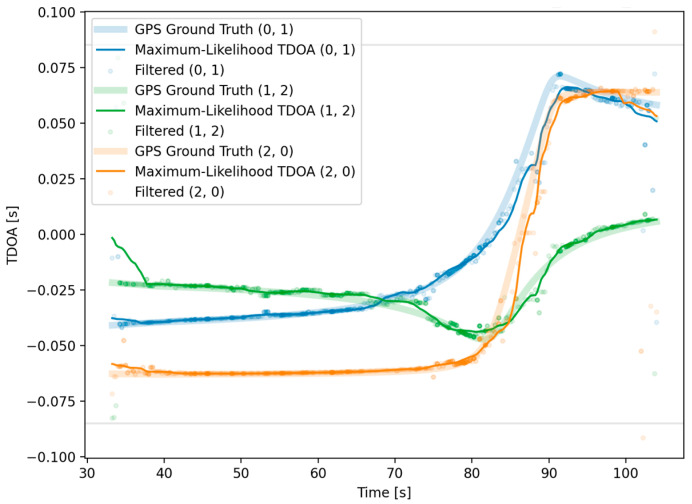
Confidence-weighted and -filtered TDOA results in high-confidence region.

**Figure 11 sensors-25-01930-f011:**
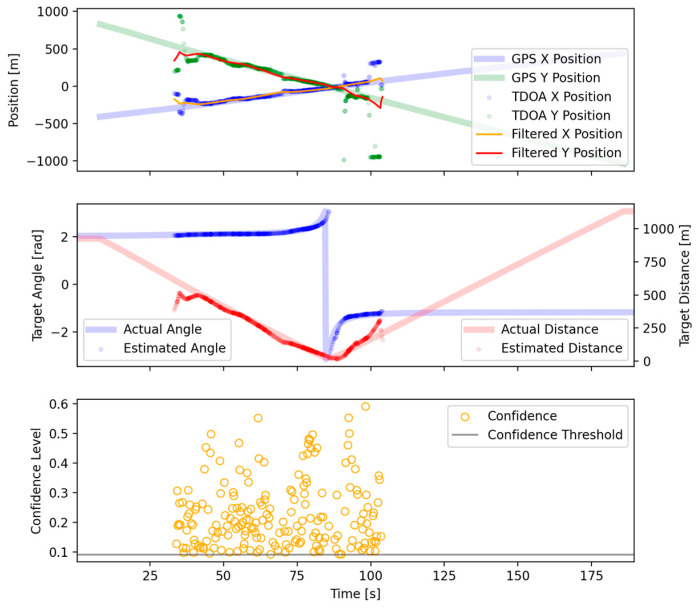
Final localization results.

## Data Availability

Datasets and code are available on GitHub: https://github.com/jeremykarst/acoustic_multilateration. Additional information and data may be made available by contacting the authors.

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
