# Peer review of "Enhancing Maritime Domain Awareness Through AI-Enabled Acoustic Buoys for Real-Time Detection and Tracking of Fast-Moving Vessels"

_sensors, 2025, doi:10.3390/s25061930_

Round 1

Reviewer 1 Report

Comments and Suggestions for Authors

This manuscript proposes a marine monitoring system based on acoustic buoys and artificial intelligence for real-time detection and tracking of fast-moving vessels. However, the manuscript falls short in terms of method description, presentation of key results, and technical details.

1.  The description of the core technical methods is overly brief, failing to elaborate on the data acquisition of the acoustic buoys, feature extraction, and the specific implementation of the AI model. This omission may hinder readers from reproducing the results or fully understanding the technical details. It is recommended to supplement the paper with a complete method description to enhance the transparency and reproducibility of the study.

2.  Figure 6 displays the acoustic waveforms and spectrograms of vessels passing through the sensor array but does not clearly explain the key information and significance. The content of the figure lacks a clear interpretation, making it difficult to understand the correlation between the vessel signal features and the research objectives. It is suggested to provide a detailed explanation of the key content of Figure 6 and offer clearer descriptions in the figure caption to help readers understand its meaning.

3.  In line 235, the term GCC-PHAT is mistakenly written as GGC-PHAT. This typographical error may cause confusion regarding the algorithms name. It is recommended to carefully check the spelling of technical terms during revision to ensure accuracy and consistency.

4.  In Section 3.1, the paper does not discuss in detail how different vessel speeds, engine types, and noise characteristics affect signal detection and localization. It is suggested to analyze these factors impact on signal features and recognition performance and explain why the selected features can effectively distinguish different types of vessels.

5.  The paper only uses Mel-Frequency Cepstral Coefficients (MFCCs) as input features and employs a simple Convolutional Neural Network (CNN) for classification. The feature extraction and fusion strategies are relatively singular and warrant further investigation. It is recommended to explore the fusion of multi-domain features (such as time-domain, frequency-domain, and time-frequency-domain) to enhance the models ability to recognize complex signals.

6.  In Section 4.1, the performance verification only presents the results of the CNN model without mentioning the recognition performance of other models discussed earlier. To demonstrate the superiority of the CNN model, it is suggested to include a comparative analysis with other models in Section 4.1 to more comprehensively showcase the advantages of the proposed method.

Author Response

Reviewer 1 Comments and Suggestions for Authors. Our replies in green highlights.

This manuscript proposes a marine monitoring system based on acoustic buoys and artificial intelligence for real-time detection and tracking of fast-moving vessels. However, the manuscript falls short in terms of method description, presentation of key results, and technical details.

  1. The description of the core technical methods is overly brief, failing to elaborate on the data acquisition of the acoustic buoys, feature extraction, and the specific implementation of the AI model. This omission may hinder readers from reproducing the results or fully understanding the technical details. It is recommended to supplement the paper with a complete method description to enhance the transparency and reproducibility of the study.

A subsection to the Materials and Methods has been added to clarify technical issues (lines 80-123). A paragraph to the data preprocessing section (lines 166-174) has been added to clarify the data manipulation procedures.

  1. Figure 6 displays the acoustic waveforms and spectrograms of vessels passing through the sensor array but does not clearly explain the key information and significance. The content of the figure lacks a clear interpretation, making it difficult to understand the correlation between the vessel signal features and the research objectives. It is suggested to provide a detailed explanation of the key content of Figure 6 and offer clearer descriptions in the figure caption to help readers understand its meaning.

Note: Figure 6 now is Figure 7. The purpose of Figure 7 is to justify our contention that our test environment is noisy and the need for careful, thorough data processing to properly identify vessel signals. The purpose of our work, to assess feasibility in such environments, was further stressed in the text associated with Figure 7 to clarify our point (lines 457-460).

  1. In line 235, the term “GCC-PHAT” is mistakenly written as “GGC-PHAT.” This typographical error may cause confusion regarding the algorithm’s name. It is recommended to carefully check the spelling of technical terms during revision to ensure accuracy and consistency.

This has been corrected on the document.

  1. In Section 3.1, the paper does not discuss in detail how different vessel speeds, engine types, and noise characteristics affect signal detection and localization. It is suggested to analyze these factors’ impact on signal features and recognition performance and explain why the selected features can effectively distinguish different types of vessels.

We did not test different vessel speed or engine configurations because our objective was to assess the effectiveness of our approach in a shallow water, near-shore environment, where ports and other facilities requiring security measures are located. Vessels in and around such locations do not have a large enough variation of speeds to provide adequate contrast in differences for producing detectable and actionable results.

  1. The paper only uses Mel-Frequency Cepstral Coefficients (MFCCs) as input features and employs a simple Convolutional Neural Network (CNN) for classification. The feature extraction and fusion strategies are relatively singular and warrant further investigation. It is recommended to explore the fusion of multi-domain features (such as time-domain, frequency-domain, and time-frequency-domain) to enhance the model’s ability to recognize complex signals.

Using complex signal implies in usage of multiple equipment and larger AI models. This will defeat the purpose of the paper to demonstrate that AI models can effectively be deployed to low-power devices. We clarified the introduction and conclusion sections to reinforce the main point of this study (lines 62-71; 573-592; 628-638).

  1. In Section 4.1, the performance verification only presents the results of the CNN model without mentioning the recognition performance of other models discussed earlier. To demonstrate the superiority of the CNN model, it is suggested to include a comparative analysis with other models in Section 4.1 to more comprehensively showcase the advantages of the proposed method.

The purpose of the paper, as discussed above, was to demonstrate an approach to detect vessels using a low-powered device. Lines 62-71, 573-592 and 628-638 were added for clarification.

Reviewer 2 Report

Comments and Suggestions for Authors

the paper is less of novelty. The methods used are too engineering and lack innovation.

Author Response

Reviewer 2 Comments and Suggestions for Authors. Our replies in green highlights.

the paper is less of novelty. The methods used are too engineering and lack innovation.

Thank you for your perspective. We have adjusted the manuscript to reflect the comments of other reviewers, which will likely address yours.

Reviewer 3 Report

Comments and Suggestions for Authors

  1. Some comments of this paper is not accurate, e.g. line 152 "Figure X".
  2. Please give more details of the AI model.
  3. The paper towards engineering application issues, the innovativeness of the AI model cannot be evaluated, and the DCLT methods lack sufficient innovation.
  4.  This paper should supplement relevant descriptions of the experimental marine environment, provide a detailed discussion on the impact of sound propagation on the TDOA algorithm, and further clarify the applicable scope of the engineering technology.

Author Response

Reviewer 3 Comments and Suggestions for Authors. Our replies in green highlights.

    1. Some comments of this paper is not accurate, e.g. line 152 "Figure X".

    The typo was adjusted accordingly

    1. Please give more details of the AI model.

    The AI model is as simple as in the description (lines 197-214) because the objective of the study was on implementation of the system on low-powered devices.

    1. The paper towards engineering application issues, the innovativeness of the AI model cannot be evaluated, and the DCLT methods lack sufficient innovation.

    See 2 above

    1.  This paper should supplement relevant descriptions of the experimental marine environment, provide a detailed discussion on the impact of sound propagation on the TDOA algorithm, and further clarify the applicable scope of the engineering technology.

    A subsection to the Materials and Methods has been added to clarify technical issues (lines 80-123). A paragraph to the data preprocessing section (lines 166-174) has been added to clarify the data manipulation procedures

Reviewer 4 Report

Comments and Suggestions for Authors

With interest and pleasure, I've got acquainted with the work devoted to the development of water area monitoring system based on acoustic buoys. The paper describes algorithms for detection and classification of fast-moving vessels, as well as prediction of their locations. The results are well illustrated, clear and of practical importance. With that, a number of remarks arise while reading the paper.

  1. A more detailed description of the hardware used, namely buoys, as well as an illustration of the network infrastructure is needed.
  2. It is necessary to present a scheme of buoys localization not in an experimental, but in an assumed state of deployment in the water area, from which the number and distance between them would be clear to solve a dedicated monitoring task.
  3. Visualizations of the feature space and training results need a better figure than a screenshot. Also, an illustration of the CNN architecture with the number and parameters of the layers, and a scheme for training the features from the data, including FFT, is also of importance.
  4. Authors should pay attention to neater formatting of the paper, including width alignment.

Author Response

Reviewer 4 Comments and Suggestions for Authors. Our replies are in green highlights.

With interest and pleasure, I've got acquainted with the work devoted to the development of water area monitoring system based on acoustic buoys. The paper describes algorithms for detection and classification of fast-moving vessels, as well as prediction of their locations. The results are well illustrated, clear and of practical importance. With that, a number of remarks arise while reading the paper.

  1. A more detailed description of the hardware used, namely buoys, as well as an illustration of the network infrastructure is needed.

A subsection to the Materials and Methods has been added to clarify technical issues (lines 80-123). A paragraph to the data preprocessing section (lines 166-174) and one describing the model architecture (lines 198-215) has been added to clarify the data manipulation procedures.

  1. It is necessary to present a scheme of buoys localization not in an experimental, but in an assumed state of deployment in the water area, from which the number and distance between them would be clear to solve a dedicated monitoring task.

We added text (lines 573-592; 628-638) to the conclusion to address this issue.

  1. Visualizations of the feature space and training results need a better figure than a screenshot. Also, an illustration of the CNN architecture with the number and parameters of the layers, and a scheme for training the features from the data, including FFT, is also of importance.

Figure 3’s information was incorporated into the text (lines 226-240) and deleted from the manuscript.

  1. Authors should pay attention to neater formatting of the paper, including width alignment.

The format of our manuscript reflects the template provided by the journal.

Reviewer 5 Report

Comments and Suggestions for Authors

1. Instead of simply referring to figures in parentheses (e.g., Figure 4), it would be preferable to incorporate them into sentences, such as "Figure 4 shows ~," for better readability and clarity.

2. On the first line of page 5, "Figure X." is mentioned. Please ensure that the correct figure number is assigned.

3. Abbreviations such as Time Difference of Arrival (TDOA) should be defined only at their first occurrence. Similarly, references should be cited only when they first appear in the text.

4. The equations at the top of page 7 and those on page 8 should be placed in a separate equations section, with equation numbers assigned for proper referencing. Additionally, all variables used in the equations, such as D0, D1, and D2, should be explicitly defined.

5. You have conducted extensive research, and the amount of information presented is substantial. However, it is difficult to grasp the overall structure of the experiment solely through text. Therefore, a framework diagram illustrating the entire experimental setup would be beneficial.

6. The subplots are referred to as "first," "last," and "middle," but using (a), (b), and (c) with individual captions would improve clarity when referencing them in the text.

7. The title mentions "Fast-moving vessels," but there is no quantitative evaluation of how the proposed method improves over existing models in detecting and tracking such vessels. A comparative analysis with prior approaches, specifically assessing performance at different speeds, would strengthen the paper.

8. The paper claims that the system operates in real-time, but it lacks quantitative evidence to support this assertion. Providing numerical data on detection and localization latency, as well as computational performance, would strengthen the real-time claim.

9. Each chapter contains an excessive amount of content and needs better organization. Additionally, the paper primarily relies on qualitative analysis; incorporating quantitative evaluations would enhance the overall rigor of the study.

Author Response

Reviewer 5 Comments and Suggestions for Authors. Our replies are in green highlights.

  1. Instead of simply referring to figures in parentheses (e.g., Figure 4), it would be preferable to incorporate them into sentences, such as "Figure 4 shows ~," for better readability and clarity.

We decided to follow standard scientific article format referring to figures after their associated findings. This keeps the text more focused.

  1. On the first line of page 5, "Figure X." is mentioned. Please ensure that the correct figure number is assigned.

The text in the manuscript has been adjusted.

  1. Abbreviations such as Time Difference of Arrival (TDOA) should be defined only at their first occurrence. Similarly, references should be cited only when they first appear in the text.

The text in the manuscript has been adjusted.

  1. The equations at the top of page 7 and those on page 8 should be placed in a separate equations section, with equation numbers assigned for proper referencing. Additionally, all variables used in the equations, such as D0, D1, and D2, should be explicitly defined.

The equations are not referred to in the document, as they are for indicating the approach used for finding optimal solutions for TDOA estimates. We added descriptions of the missing variables, as suggested.

  1. You have conducted extensive research, and the amount of information presented is substantial. However, it is difficult to grasp the overall structure of the experiment solely through text. Therefore, a framework diagram illustrating the entire experimental setup would be beneficial.

We added a figure (Figure 2) clarifying the sequence of steps from data gathering to deployment.

  1. The subplots are referred to as "first," "last," and "middle," but using (a), (b), and (c) with individual captions would improve clarity when referencing them in the text.

We added subplot labels to figures 7 and 9 and adjusted associated text.

  1. The title mentions "Fast-moving vessels," but there is no quantitative evaluation of how the proposed method improves over existing models in detecting and tracking such vessels. A comparative analysis with prior approaches, specifically assessing performance at different speeds, would strengthen the paper.

The purpose of the paper was to demonstrate an approach to detect vessels using a low-powered device (our contribution). Other methods might be more effective in detection, but that effectiveness comes at a higher price and requires additional logistics. We did not draw any comparisons, as they would not be based on equal standards. Lines 62-71, 573-592 and 628-638 were added for clarification.

  1. The paper claims that the system operates in real-time, but it lacks quantitative evidence to support this assertion. Providing numerical data on detection and localization latency, as well as computational performance, would strengthen the real-time claim.

We added information on latency to the materials and methods section (lines 100-102)

  1. Each chapter contains an excessive amount of content and needs better organization. Additionally, the paper primarily relies on qualitative analysis; incorporating quantitative evaluations would enhance the overall rigor of the study.

The introduction, materials and methods, and the conclusion sections have been modified and extended to solidify the objectives, detail methodological approaches, and refine conclusions.

Round 2

Reviewer 1 Report

Comments and Suggestions for Authors

After the revision, the quality of the manuscript has been significantly improved. There are no more questions.

Author Response

Thank you for your input and comments throughout this process.

Reviewer 2 Report

Comments and Suggestions for Authors

The revised manuscript presents a comprehensive study on enhancing maritime domain awareness (MDA) using AI-enabled acoustic buoys. The research is timely and relevant, addressing critical challenges in maritime security and environmental protection. The integration of AI with acoustic sensing systems offers a promising solution for real-time detection, classification, localization, and tracking (DCLT) of vessel activity, particularly in scenarios where traditional monitoring systems face limitations.

The manuscript is well-structured, with clear sections that guide the reader through the research process. The introduction effectively establishes the importance of MDA and the limitations of existing technologies, setting the stage for the innovative approach proposed. The materials and methods section provides detailed technical specifications of the OpenEar™ system, data collection protocols, and the AI model architecture, allowing for replication of the study. The results are presented with appropriate visualizations and analyses, supporting the claims made by the authors. The discussion interprets the findings in the context of previous research and highlights the significance of the results for MDA applications.

Comments on the Quality of English Language

The introduction could benefit from a more detailed comparison between the proposed AI-enabled acoustic buoy system and other emerging technologies for MDA. This would help position the study within the broader landscape of maritime surveillance solutions.

The equipment description is thorough, but additional information about the durability and maintenance requirements of the OpenEar™ system in different environmental conditions would be valuable.

The data collection section could include more details about the specific challenges encountered during field deployment and how they were addressed.

The AI model section is technically sound, but a more in-depth explanation of the hyperparameters and optimization techniques used during training would enhance transparency.

The presentation of results is clear, but including statistical analyses (e.g., confidence intervals, significance tests) would strengthen the validity of the findings.

The discussion of localization accuracy could be expanded to include a comparison with other TDOA-based methods and an analysis of the factors contributing to the observed errors.

The authors should elaborate on the potential limitations of the current approach, such as its dependence on specific environmental conditions or vessel types.

A more detailed discussion of the scalability of the system, including cost considerations and infrastructure requirements for larger-scale deployments, would be beneficial.

The conclusion effectively summarizes the main findings and their implications. However, it could benefit from a clearer statement of the study's contributions to the field and future research directions.

Author Response

Please note that our replies are in green. The black font is the original text from the reviewer.

The revised manuscript presents a comprehensive study on enhancing maritime domain awareness (MDA) using AI-enabled acoustic buoys. The research is timely and relevant, addressing critical challenges in maritime security and environmental protection. The integration of AI with acoustic sensing systems offers a promising solution for real-time detection, classification, localization, and tracking (DCLT) of vessel activity, particularly in scenarios where traditional monitoring systems face limitations.

The manuscript is well-structured, with clear sections that guide the reader through the research process. The introduction effectively establishes the importance of MDA and the limitations of existing technologies, setting the stage for the innovative approach proposed. The materials and methods section provides detailed technical specifications of the OpenEar™ system, data collection protocols, and the AI model architecture, allowing for replication of the study. The results are presented with appropriate visualizations and analyses, supporting the claims made by the authors. The discussion interprets the findings in the context of previous research and highlights the significance of the results for MDA applications.

Comments on the Quality of English Language

The introduction could benefit from a more detailed comparison between the proposed AI-enabled acoustic buoy system and other emerging technologies for MDA. This would help position the study within the broader landscape of maritime surveillance solutions.

We opted to detail the suggested comparison in the conclusions section, as the reader at that point would be more familiar with the characteristics of our system.

The equipment description is thorough, but additional information about the durability and maintenance requirements of the OpenEar™ system in different environmental conditions would be valuable.

We refrained from providing details about durability because our experimental setting included only one location. Extrapolations to different conditions would, therefore, be only conjectural.

The data collection section could include more details about the specific challenges encountered during field deployment and how they were addressed.

We added more details on challenges faced during our study and on typical challenges when deploying and using passive acoustic devices similar to ours (lines 616-627 but also see 629-639 of previous version).

The AI model section is technically sound, but a more in-depth explanation of the hyperparameters and optimization techniques used during training would enhance transparency.

We added additional details for the model on lines 276-282.

The presentation of results is clear, but including statistical analyses (e.g., confidence intervals, significance tests) would strengthen the validity of the findings.

Our experimental setting did not lend itself to replication. We gathered sufficient acoustic data to generate a CNN model with strong metrics but not for providing robust statistical inference.

The discussion of localization accuracy could be expanded to include a comparison with other TDOA-based methods and an analysis of the factors contributing to the observed errors.

There are many variations of TDOA which apply to more sophisticated microphone arrays, or with direction of arrival information, but they all require different hardware configuration than was used here. We feel that the TDOA techniques employed are sound from a mathematical point of view, with only potential improvements if a different optimization solver, offering only minimal benefits.

The authors should elaborate on the potential limitations of the current approach, such as its dependence on specific environmental conditions or vessel types.

As above (see lines 616-627).

A more detailed discussion of the scalability of the system, including cost considerations and infrastructure requirements for larger-scale deployments, would be beneficial.

We added text to this effect on lines 82-96. The text elaborates on system details, including scalability.

The conclusion effectively summarizes the main findings and their implications. However, it could benefit from a clearer statement of the study's contributions to the field and future research directions.

We added text further highlighting the applicability of our study (lines 650-667).

Reviewer 4 Report

Comments and Suggestions for Authors

I thank the authors for their work on the manuscript and the reviewer's comments. The work has improved considerably, but there are still a number of issues to be addressed.

1. Fig. 3 should be enlarged to provide better readability.

2. In the buoy application recommendations, numerical estimates, hypothetical or based on research results, are required, such as the density of buoys and the depth of the coastal zone that can be covered by a single row of buoys.

Author Response

As for reviewer 2, our replies are in green and the original suggestions in black.

I thank the authors for their work on the manuscript and the reviewer's comments. The work has improved considerably, but there are still a number of issues to be addressed.

  1. Fig. 3 should be enlarged to provide better readability.

We have enlarged Figure 3

  1. In the buoy application recommendations, numerical estimates, hypothetical or based on research results, are required, such as the density of buoys and the depth of the coastal zone that can be covered by a single row of buoys.

We added a separate paragraph including the physical details of the system, together with associated characteristics (lines 88-96).

Reviewer 5 Report

Comments and Suggestions for Authors

I have confirmed that the revisions effectively incorporate the review comments, enhancing the clarity and coherence of the paper.

Author Response

(The authors gave the same response as above.)
